# Situating the Father: Strengthening Interdisciplinary Collaborations between Sociology, History and the Emerging POHaD Paradigm

**DOI:** 10.3390/nu14193884

**Published:** 2022-09-20

**Authors:** Christopher Mayes, Elsher Lawson-Boyd, Maurizio Meloni

**Affiliations:** Alfred Deakin Institute for Citizenship and Globalisation, Deakin University, 75 Pigdons Rd, Geelong 3216, Australia

**Keywords:** epigenetics, history, sociology of food, gender, nutrition, DOHaD, POHaD

## Abstract

(1) Background: Albeit the main focus remains largely on mothers, in recent years Developmental Origins of Health and Disease (DOHaD) scientists, including epigeneticists, have started to examine how a father’s environment affects disease risk in children and argued that more attention needs to be given to father’s health-related behaviors for their influence on offspring at preconception (i.e., sperm health) as well as paternal lifestyle influences over the first 1000 days. This research ushers in a new paternal origins of health and disease (POHaD) paradigm and is considered a welcome equalization to the overemphasis on maternal influences. Epigeneticists are excited by the possibilities of the POHaD paradigm but are also cautious about how to interpret data and avoid biased impression of socio-biological reality. (2) Methods: We review sociological and historical literatures on the intersection of gender, food and diet across different social and historical contexts to enrich our understanding of the father; (3) Results: Sociological and historical research on family food practices and diet show that there are no “fathers” in the abstract or vacuum, but they are differently classed, racialized and exist in socially stratified situations where choices may be constrained or unavailable. This confirms that epigeneticists researching POHaD need to be cautious in interpreting paternal and maternal dietary influences on offspring health; (4) Conclusions: We suggest that interdisciplinary approach to this new paradigm, which draws on sociology, history and public health, can help provide the social and historical context for interpreting and critically understanding paternal lifestyles and influences on offspring health.

## 1. Introduction

Epigenetics has traditionally focused on mothers and the maternal environment as possible site of intervention [1,2,3,4,5,6,7]. In some of the most cited publications in epigenetics (for instance, the agouti mouse study [8] and Meaney and colleagues’ rat experiments [9,10,11]), modification is invariably “introduced via the behavior or physiology of the mother”, a view that is increasingly understood as “an intensified space for the introduction of epigenetic perturbations in development” [12]. This is of course hardly new, considering how special attention to material behaviour in relation to the offspring’s health and development is heavily documented in history of medicine, and more recently development origins of health and disease (DOHaD) [7,13,14]. Evolutionary theory has also often privileged the mother-foetus and newborn dyad given its importance for offspring development [15] and survival particularly in mammals [16], although a wider analysis of biparental care has highlighted the benefits and evolutionary advantage of synergistic parenting [17].

Partly as a response to the concern that an uncritical reception of DOHaD and epigenetics may augur further control of the maternal body, in recent years epigenetic researchers have investigated how a father’s environment affects disease risk in children and argued that more attention needs to be given to father’s health-related behaviors for their influence on offspring at preconception (i.e., sperm health) as well as paternal lifestyle influences over the first 1000 days. This research ushers in a new paternal origins of health and disease (POHaD) paradigm and is lauded by many as a welcome equalization to the overemphasis on maternal influences [18,19,20] and an expansion of epigenetic and intergenerational research beyond the intrauterine environment into areas traditionally neglected (such as fathers exposure to drugs rather than mothers, etc.) [21,22]. POHaD has emerged recently in the light of a wealth of new research on the intergenerational transmission of paternal environmental information in epidemiology, environmental toxicology, animal models, and molecular biology over this last decade [19,20,21,23,24,25,26,27,28,29,30,31]. However, in examining these developments through the lenses of history and sociology of health we suggest that there a number of social implications and historical precedents of the POHaD paradigm that epigeneticists should consider.

The novelty of POHaD paradigm can be usefully contextualised in research conducted in history and sociology of health. While POHaD researchers note an overemphasis on the mother as determining offspring health, there is a long history of attention given to the father too, in medical and biological approaches. For example, dietary guides during the Renaissance were written by men for other men and rarely addressed women’s needs. Historically, a father’s ability to control their appetite was also considered a sign of the ability to control their body, family, and by extension govern the city. Attention to fathers’ behaviour, habit, and lifestyle for the optimization of reproduction, particularly in relation to alcoholism as “racial poison”, was also part of different strands of the eugenic and the temperance movement in the early twenty century, including the notion of “spermatic infection” that may “injure posterity” [14,32,33]. More recently (i.e., mid-20th C onwards), sociologists have focused on the gendered politics of health by critically examining the influence of the father’s dietary preferences on the family meal, and subsequently family health. Public health programs, particularly in relation to obesity and other non-communicable diet related diseases, have used this research to focus on fathers as a strategic target for changing the diets of the entire family [34]. Furthermore, advocates for the heteronormative nuclear family as the rightful foundation of a stable social order also centre the importance of the father during family mealtimes.

Epigeneticists are excited by the possibility of a new POHaD paradigm but are also cautious about how to interpret data and avoid biased impression of socio-biological reality. This caution is warranted given the way the fathers’ diet, habits (including drinking) and role during mealtimes has been used in various social and political schemes in the past and present. In this paper we outline how sociology and history can provide a broader context for POHaD as well as indicate some of the potential socio-political pitfalls. We show that there are no “fathers” in the abstract or vacuum, but they are differently classed, racialized and exist in socially stratified situations where choices may be constrained or unavailable. We suggest that interdisciplinary approaches to this new paradigm, which draws on sociology, history and public health, can help provide the social and historical context for interpreting and understanding paternal lifestyles and influences on health in a critical and self-reflexive way.

## 2. Materials and Methods

This research article uses methods from the humanities: sociological and conceptual analysis. This approach does not readily conform to scientific research design and ideals of reproducibility. While sociological research can use explicit qualitative and quantitative methods to investigate social phenomenon, this article is not based on empirical research. Rather, it contributes to current debates about POHaD by building on the research of previous sociological studies of food and family life, as well as historical archival work on food and diet. In bringing sociological and historical research into conversation with current POHaD research we aim to provide a critical contribution that may be helpful for translation of scientific research into dietary practices and food cultures.

First, we outline our understanding of the epigenetics research on POHaD, showing the ways in which paternal diet and their health effects are described. Second, we provide an overview of how the fathers’ diet and eating behaviours have been understood in different historical periods as influencing the health of offspring and broader socio-political implications. Third, we show how sociologists of health have critically examined the gendered politics of family health and meals. Finally, we discuss the relevant lessons for the POHaD paradigm that can be learnt from historical and sociological literatures.

## 3. Results

### 3.1. Overview of Epigenetics Research on POHaD

This is a brief overview of recent studies that examine correlations between paternal diet and the epigenetic signatures. To simplify, the studies and reviews included here collectively reflect an assertion that epigenetic studies will strengthen public health policies that target consumption patterns and dietary choice of fathers (fathers to-be especially). While the lifestyles and behaviors of women are commonly targeted as the profound sculptor of their offspring’s future health trajectory [35] this overview reflects a growing intensity of paternal impressions. In other words, it demonstrates an overt focus on sperm health and dietary choices at preconception, thus providing epigenetic grounds for interventions in the lives of fathers and fathers to-be.

Epigenetic studies entail a concern for natures of heredity—what biological dispositions acquired through “behavioural choices” and “lifestyles” are passed onto following generations—with dietary choice considered a major contributing factor. Fundamental to epigenetics is the idea that “environments” not only act as mediators of gene expression, but as exposures, which assumes anything from food to green-house gas to stress as biologically noxious and transmissible when ingested, consumed and absorbed. While intergenerational research in epigenetics—specifically in humans—remains thin and hotly contested, researchers in the field argue for an intergenerational context in which to imagine, regulate and monitor health outcomes of populations. As US sociologist Hannah Landecker in a widely influential article has argued: “The experimental image of human life generated in rodent models (…) generates concepts of food as a form of molecular exposure. This scientific discourse has profound implications for how food is perceived, manufactured and regulated, as well as for social theories and analyses of the social body that have a long history of imbrication with scientific models of metabolism” [36] (p. 167, see also in Indian context, [37]).

Despite the attention to environments and intergenerational effects, which suggest a broad scope of influences, the focus of epigenetics has been predominantly on mothers and the maternal environment [1,2,3,4,5,6,11]. Until recently, paternal epigenetic inheritance has received scant attention. For example, it is proposed that paternal obesity, as a result of an excessive consumption of calorie dense, poor-quality foods, can predict adverse health consequences for the offspring’ [38] (p. 132). Yet, it should be noted that these conclusions are primarily based on rodent studies and researchers have not hesitated in lamenting the translational opportunities for rodent models [38].

In the context of paternal dietary studies, researchers control the diet of their rodent subjects portionally (i.e., feeding rodents higher portions of saturated fats and refined sugars). While most studies use rodents as the model, it is not yet known whether paternal obesity in humans also impacts on the reproductive health of children and grandchildren [38]. Despites these limitations rodent studies have been used to examine: (1) how specific dietary and activity patterns raise the risk of obesity, (2) how obesity influences epigenetic profiles, and (3) how these epigenetic profiles affect the integrity of sperm and the subsequent health status of future generations (more specifically, how epigenetic modifications may raise the risk of obesity and metabolic disorders in offspring).

While it may be obvious that rodent models clearly do not encompass the intense complexity of food environments in modern post-industrial worlds, they illustrate and reinforce public health communications such as self-regulation of diet, managing stress, regular exercise, etc. Yet, the POHaD also emphasises what men choose to eat will not only affect their own health, but their lineage (i.e., children and grandchildren—the kin they will most likely live to see). A growing body of evidence from preclinical studies in rodents suggest that the progeny of high-fat diet males undergo abnormal embryonic development [24,25,39]. This includes reduction in the cleavage rate of zygotes, a significant delay in preimplantation development, and reduced implantation, leading to markedly fewer embryos descending from HFD-fed males reaching the blastocyst stage compared with controls. Of those embryos that reached the blastocyst stage, blastocyst cell numbers were decreased in embryos sired by HFD-fed males, which is indicative of reduced viability. Moreover, blastocysts from obese males showed a greater proportion of cells that were apoptotic. Strikingly, recent data from a human fertility center revealed prolonged durations of embryonic cell cycles, with delayed cleavage intervals in the embryos fertilized by obese men compared with those of lean men. Such delays have been associated with a reduced rate of blastocyst expansion and implantation. A similar phenomenon in humans has been observed with regard to advanced paternal age. Mechanisms that could be responsible for this delay include DNA damage in spermatozoa, as noted previously [40,41,42].

Furthermore, small changes in sperm RNAs have the potential to produce long-term metabolic consequences in offspring by regulating embryo mRNA at an early stage through altered gene expression within metabolic pathways [43]. Moreover, the cord blood from newborn offspring of obese fathers showed distinct hypomethylation at the insulin-like growth factor 2 (IGF2) differentially methylated regions (DMRs) [19]. As changes in the methylation of the IGF2 gene have been linked to chronic adverse health outcomes and an increased risk of developing cancer, the authors suggested that obesity-associated hormone imbalances might mediate incomplete or disorganized methylation of the IGF2 gene during spermatogenesis [19].

Fathers’ risk factors, including high body fat mass, overweight, altered gut microbiota, increased estrogen levels, among others, are said to have effects on offspring. These effects can include reduced birth weight, increased adiposity and obesity risk, altered neonatal brain function connectivity, metabolic dysfunction, altered gut microbiota, subfertility, increased anxiety and depressive like phenotypes, and cognitive impairments [38], p. 132. Significantly, research in the POHaD paradigm also notes an effect on sperm (such as decreased motility, impaired mitochondria, DNA hypomethylation, and altered RNA signatures) and embryo (reduced fertilisation, delayed preimplantation development, and reduced implantation) [38].

Crucially, this overview highlights the prevalence of the discourse of prevention, whereby the diets and lifestyles choices of future fathers will ultimately play a major role in sculpting subsequent generations. However, this is not without precedence has can be seen from the historical and sociological literatures.

### 3.2. Historical Focus on Fathers’ Diets

Historians of food and health have conducted important research useful for contextualising the father’s diet. A detailed history of the way a fathers’ diet has been understood and the different and complex perception of the link between “obesity” and disease or gluttony and lack of moderation and intemperance is clearly not feasible in this short article. Even less it is possible to look at the incredibly rich connection of medicine and herbs (but also perfumes, flowers), food and, ways of cooking it, from the Hippocratic On Regimen (Fifth and fourth century BCE), to Galen’s (129–216? CE) work on dietetics, such as his On the Properties of Foodstuffs, to Ibn Sina’s influential canon of medicine (a key medical text in European Universities until the sixteenth century) (980–1037) to the Renaissance flourishing of nutritional manuals [44,45,46,47,48,49]. To offer just a summary, a deep anxiety toward food as a way of life is ubiquitous in the pre- and early modern experience in the context of what after Galen are called the six res non-naturales, that is all factors that could shape humoral balance, and had hence to be carefully governed: food, certainly, but also, sleeping, emotions, exercise, airs, evacuations [13,50]. 

To deal with a more tractable aspect of this too wide debate over the interactions among food, bodies and environments in the pre- and early modern period, we want to focus instead on a very specific, mechanistic aspect. What was the connection, before the rise of genetics, between food and reproduction, including not just sterility or fertility but the quality of the offspring and its health, that is, in a modern word “heredity”? Without aiming to homogenise different historical experiences, it is fair to say that in the absence of a clear division between germplasm and somatic cells, which will be clearly introduced only by August Weismann at the end of the nineteenth century, the relationship between food and semen and hence reproduction in Hippocratic and Galenic medicine was particularly blurred and difficult to disentangle. In the theory of pangenesis (direct communication between somatic and germ cells) embraced by the premodern medical tradition although not shared by all natural philosophers (for instance Aristotle), the “semen comes from all parts of the body” (Hippocrates, On Airs, Waters and Places) [51] (p. 161). It is, in summary, a digested (or concocted) food, like a distillation process at the end of which the spermatic substance, in both sexes (again not for Aristotle) is released. The “fungibility” and physiological affinity [52] between different body fluids—blood, milk, semen—and functions like digestion and reproduction in the premodern humoral experience generate a number of interesting problems and sometimes true anxieties about the quality of offspring. Specific or too intense types of food could lead to uncontrolled sexual desire or, vice versa, not enough digested food could compromise the quality of the semen and hence the health of the progeny. Given that “production of sperm is merely the last step of the entire digestive process, and it is generated directly from an excess of nutritive material remaining after the body has been nourished” [44] recommendations to digest food properly before having sex are frequent in this context [52,53] (p. 50). The experience of the body in premodern medicine is very much about self-control, moderation, and a careful management of all possible environmental exposures which may influence or perturb physiological balance.

Islamic medicine in the medieval period built on and expanded the scope of Greek humoral writings. For instance, it was recommended that having a happy and joyful mind during copulation and avoiding specific winds may produce more lazy or delicate children [54]. Food rules were also important for reproduction. Parents who consumed “unlawful foods” were at risk of producing offspring with weak complexion or a corrupted soul. Interestingly in many of these premodern cases (including the wider topic of maternal impression [55]) what can be read into a child’s body or mind is not just a random misfortune but is taken as a direct effect of some misbehaviour of the parents. While premodern and early modern medicine did recognize limitations to the power of food to change the existing balance and constitution of different patients, it also inspired substantial anxieties about the effects of change in food and diet over the health and characteristics of whole groups not just individuals [56] (p. 50). For instance, as documented in later colonial contexts, the extreme permeability of the semen to change in diet and location operates as a channel through which new environments can leave direct marks on the bodies of human groups [13] (p. 62). Not only, as in in Greek and Roman military literature “soft places” produce “soft people” but more specifically different foods eaten (and waters drunk) at different locales shaped the semen differently contributing to physiological and even racial changes in the progenies [51].

Similar anxieties about the stability of bodily boundaries and the risk of racial reversibility became increasingly important at the time of the first colonial movements to the New World. In *The Body of the Conquistador* (2012), historian Rebecca Earle describes the complex micropolitics of food that Spanish colonizers promoted in the new colonies [57]. If existing knowledge in medical humoralism—implied that with a change of place (different food and waters, but also stars and winds) also people would gradually change, then the Spanish body would mutate in the New World. This is why a specific attention to diet “more than any other factor” was used to maintain a separation between Spanish and Amerindian bodies. Spaniards therefore imported bread, wine, oil, and crops from the mother land not so much for “culinary nostalgia” but for protecting the integrity and stability of the colonizers’ body [57].

As Earle writes, “Concern that exposure to different foods and environments spoke directly to Spanish worries about the physical integrity of their bodies, and about the maintenance or dissolution of the most fundamental of colonial divisions: that between the bodies of the colonisers and the colonized” [57] (p. 183). As often in the long history of biological plasticity, the permeability of racial characteristics to food environments could generate different and often competing political agendas and moral dilemmas. Views of degeneration or regeneration, as well as politics of separation or friendship, could be equally applied to the porous body of humoralism. Columbus, for instance, supported the importance of Spanish food to protect the colonizer’s body and restore health among settlers. He insisted on the importance of old world varieties in the New World, importing legumes and grapes, sugar cane and stock animals [57] (p. 83). However, this discourse could easily be inverted to promote a more optimistic equality between colonizers and colonized. In his defence of Amerindians, the Dominican friar Bartolomé de las Casas for instance also attributed key transformative value to a change in diet. He not only believed that given their brotherhood and common descent from Adam and Eve, the only explanation of the physical differences between colonized and colonizers would food. He also claimed that a proper usage of food may reduce any visible difference between settlers and colonized bodies: “with the right foods … Amerindians might perhaps come to acquire a European constitution” [57] (p. 3). Claims of a fundamental inferiority of native bodies were therefore unjustified [58,59].

The point of this admittedly brief and idiosyncratic history is to show that past approaches to diet were also charged with deep medical and political significance. Albeit in different institutional, economic and moral contexts, the notion that diet and environments were thought to have transformative powers and shape the bodies, health, and physical appearance of individuals and populations runs like a thread across different epochs. Additionally, dietary knowledge and guidance was particularly concerned with the determining effects of fathers’ diets. Dietary guides were almost entirely directed at male audiences, and recommendations rarely specify women’s needs, and if they did it was only by contrast to males [44].

### 3.3. Sociology of Fathers’ Dietary Preferences and Impact on Family Health

Although as we have noted there have been shifting attention to the role of fathers or mothers (for instance in early twentieth century eugenics), sociologists, like epigeneticists, acknowledge that the mother’s influence on the health of offspring has been the overwhelming focus of health policies and medical discourse over these last few decades [33,60]. It is important to note that food practices vary across cultures and societies, as well as within them. As the preceding section made clear, food practices also vary across time and can be disrupted, transformed and repurposed. The POHaD paradigm may be part of a process that disrupts and transforms current understanding of food, diets, and intergenerational health. However, drawing on the sociological literature that examines food practices in Western liberal societies we show how gender and patriarchal power relations shape diets, mealtimes, and child health.

The preparation, serving, and consumption of food in the domestic environment has been studied by sociologists since at least the middle of the twentieth century, particularly the role of class and gender. It is within the family that “status and power differences according to gender can be reflected in the distribution of food (the qualities, quantities, and manner of serving food)” [47,61]. Meat consumption is a useful illustration of gendered differences in the family. In post-industrial western societies meat has become associated with economic wealth and male dietary preferences with women giving “priority to male preferences at the expense of their own tastes” [47,61]. Male tastes are not only prioritised, but they are often given greater quantity of meat partly due to nutritional beliefs that men need more protein than women [62]. There is a vast literature on the gendered politics of meat and the family meal, with a number of studies looking at how a fathers’ meat consumption shapes the meat consumption of their children leading to associated disease risks [63,64,65]. The relevant point here is that socially constructed male food tastes and satiety needs are prioritised over the women who are tasked with the responsibility to prepare the meal.

Sociologists of food note the importance of the family meal in developing and performing social roles. These roles include the non-neutral domestic acts of who gives and who receives food, as well as the etiquette and the way one eats [48,61]. This can include specifics of using utensils in the appropriate way that reflects class and civility [66]. The way one eats and ability to demonstrate control over one’s appetite has wider significance. The way one performs their role can reflect moral and political qualities [35]. For instance, as mentioned above, a father’s ability to control his appetite and his household has been interpreted as a reflection of his ability to perform roles in society and public office.

Furthermore, the family meal serves as a means of socialisation into domestic duties and spaces [67]. Sociologists studying women’s magazines in the second half of the twentieth century note the division of domestic spaces, citing the consistent “separation of the mother and daughter from the father and or male children in the kitchen space” which “clearly demarcates who carries out the food work, and exactly how it should be done" [68] (p. 143). This historical sociological analysis is consistent with more recent work on the division of labour in the family which maintains that the “overall responsibility for domestic affairs falls to the woman of the house” [61] (p. 54).

While public health practitioners and epigenetics researchers may feel uncomfortable with these gendered domestic arrangements, their work arguably reinforces these arrangements by adding the health of children and future children to the list of the mothers’ responsibilities [69]. Yet, the mother’s responsibility historically and in the present has been to serve the dietary interests of the male members of the household. While the male members of the home may be absent from the kitchen, their preferences, particularly the fathers’, has a determining effect of the food prepared and served, and thereby influencing the diets and health of their children. For example, researchers in the United States found that in prioritising fathers’ taste and culinary desires “fathers’ can undermine mothers’ attempts at healthy eating within families” [70] (p.99). Sociologists have also examined how public health researchers have been concerned by the influence of the father in shaping the unhealthy eating behaviours of their children, particularly in relation to obesity [70,71,72]. Like the POHaD, public health researchers have considered the father as the “forgotten parent” in strategies to prevent childhood obesity [34,73]. The father may be “forgotten” but as sociological studies have shown the father has significant influence over the domestic sphere and dietary practices of the home. A challenge for both epigenetics and public health researchers is how to address the fathers’ influence on family health and dietary preferences without naturalising or reinforcing it.

Thus far, we have drawn on sociological literature focusing on class and gender analysis of the family and food. In addition to this literature, recent studies in the sociology of science have critically examined the gendered politics of epigenetics, including a specific gendered figuration of paternal influences [13,32,74,75,76,77,78]. Science does not occur in a vacuum and, as we know from the unsettling history of the entanglement of science and racism, sexism may operate also in science at an unconscious level, bringing to light only certain questions and research designs while sidelining alternative approaches that might foster greater equality [7,50].

The relevant point here is that POHaD may succumb to a similar hijacking. Despite the intentions of epigenetic researchers to broaden the focus from the mother to include the father, scientists need to remain open to the complex local contexts in which different politics of food based on gender imbalance may occur. Inattention to these contexts when translating research may undermine steps toward a fairer society by reinforcing the role of women as responsible for the dietary health of the family or stimatising fathers and future fathers for behaviours that are in fact determined by wider societal and economic causes [74]. Conducting research and translating into context without appreciating these dynamics may in fact serve to replicate or exacerbate the transmission of existing inequality. Moreover, such paternal focus may actually reinforce a longer historical tendency to privilege a male audience in treatises on food [44].

## 4. Discussion

While there are obviously novel aspects of the POHaD paradigm, it is also part of a longer history of attention given to father’s diet habit and behavior for the optimization of reproduction [14,79]. This history, as well as recent sociological analyses of the father’s diet and the family meal provide useful context for POHaD. It also provides salutary lessons.

The family meal is loaded with normative claims about natural social order, subservient roles of women, and male authority. Most sociologists do not condone these mealtime dynamics, rather they describe them in order to critically evaluate how these practices reflect gendered and patriarchal ordering of society in the domestic sphere. However, there are researchers and pundits who examine the family meal for very different purposes. There are those who believe “the West” or the United States is in moral, social, economic, and health decline due a “crisis of masculinity” [80]. Obesity, for example, is considered a physical manifestation of a much wider problem, namely the decline of (white) American values, which has a similar cause with slipping family values. From this perspective, the father needs to be reinstated at the head of the family table, with a wife serving a meal of animal protein and obedient children respectfully seated.

The brief sketch of historical and sociological understanding of diet, families and gender show that “men”, “fathers”, and their diets do not exist in a vacuum. While a shift in focus that balances what happens within the intrauterine environment with broader external factors, including paternal exposures, is certainly beneficial, current debates within western liberal societies about the politics of the family meal could result in POHaD reinforcing heteronormative and patriarchal ideals of food preparation and consumption [81,82]. The politics of the family meal with the father at the centre of it has a long and troubled history. As noted, in antiquity the father’s eating practices indicated his ability to control himself, the family, and the city. In more recent years, sociologists of food have noted the centrality of the family meal to sustain patriarchal and heteronormative values. Epigenetics may wish to bracket these concerns to only focus on health outcomes, but to disentangle a science of food from its wider societal effects may not be entirely possible. Like genes or bodies, health, food, and fathers are socially, culturally, and historically situated and shaped.

We suggest that epigenetics research in general and POHaD specifically needs an interdisciplinary approach to food and health that incorporates both natural and social sciences as well as humanities. To the extent that epigenetics brings into its research questions topics such as gender, diets, culture, and history, and in general social contexts, the need for a cross-disciplinary dialogue becomes unavoidable. For example, Müller et al. outline the importance and value of an interdisciplinary approach to epigenetics research questions [83,84]. The social cannot be taken out of food or health, nor can be bracketed the long term historical background that gives significance to contemporary findings and shape and organize research questions in certain directions rather than others. This situation “opens up novel opportunities for collaboration between researchers in biology, the social sciences, and the humanities” [83] (p. 1678). The shift among epigeneticists to focus on men may be taken by itself as an illustration of the inevitable entanglement of social and biological values, given how this shift was also in part influenced by social pressures and important feminist critiques of DOHaD research [85]. 

## 5. Conclusions

While a neat division between genetic and environmental causes might have suggested in the past the notion of a hiatus between biological and social disciplines, the biosocial entanglement of bodies and cultures that emerges today with epigenetics requires the establishment of “trading zones” [86] where different forms of expertise can be heard and collaborate to promote a science that is at the same time accurate and aware of its social implications and historical context.

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
