# Peer review of "Situating the Father: Strengthening Interdisciplinary Collaborations between Sociology, History and the Emerging POHaD Paradigm"

_nutrients, 2022, doi:10.3390/nu14193884_

Round 1

Reviewer 1 Report

This paper describes the role of a father's environment as a risk factor for disease in children. I believe that the manuscript is well written overall and the data align with the overall aim of this paper. I have a few minor comments for this paper.

1) Section 3.3. Sociology of father's food and health. While this section is well written, it appears that more emphasis is given on the role of the father as a patriach of the household than how the diet of the father actually influences the next generation. It has been mentioned at some point in this section but it does not convey the message as mentioned in the heading of the section. Maybe this section could be written to tailor the section heading.

Author Response

·    We have slightly changed the title of the section to “Sociology of fathers’ dietary preferences and impact on family health”. This better reflects the points being made in this section.

·    We have also added sentences and additional references (see highlighted) throughout making a stronger link between fathers’ dietary preferences and the preferences and health of offspring.

·    Furthermore, the reason for emphasising the role of the father as patriarch in the household is to highlight the social context into which POHaD messaging maybe entering and unwittingly justifying the central of the father in areas beyond diet and nutrition.

Reviewer 2 Report

Mayes  et al present an interesting and relevant article, where they aim at situating the father in the context of the paternal origins of health and disease, both from a historical and sociological perspective. This is of great merit, as in addition to biological aspects, political and sociological ones can also influence fathers and fathers to be impact on their descendants health. Of notice, very few articles are available in the POHaD literature takinh such an interdisciplinary approach, linking biology with social sciences.

Here we would like to highlight some points that we consider may improve the quality of the article.

1. Authors mention that epigeneticists are mainly interested in the POHaD paradigm. In our view, it would be better to say that Developmental Origins of Health and Disease (DOHaD) scientists, including epigeneticists, have been working on the influence of fathers on desendants health. Clearly, scientists working on the POHaD field are not limited to epigeneticists.

2. On lines 11 and 37, authors mention that POHaD approaches are or are considered a wellcome corrective to the overemphasis on maternal studies in DOHaD. We suggest substituting "corrective" to equalization. Furthermore, authors should better contextualize in the introduction why there has been an overemphasis on the mother in the DOHaD context, both from a sociological/historical and biological perspective as well. One may argue that from a biological perspective that the dyad mother-fetus and mother-newborn presents the closest association from a developmental perspective...

3. line 31. Please define DOHaD.

4. It would be important to mention in the introduction how father´s behaviours could impact descendant´s health over the developmental window of the first 1000 days. It seems that from what has been presented, that because men can have a profound impact on family meal dynamics, this could be an example.

5. Authors should better contextualize the emergence of the POHaD paradigm by citing also biological reasons why at some point over the last decade interest started to be directed to the role fathers can have on theior descendants health in the context of DOHaD studies.

6. Lines 22 and 66. It would important to mention "influences on offsping health".

7. line 87. define where epigenetic signatures are found (i.e. sperm, offspring tisues?).

8. In section 3.1, it would be important to present broader examples of epigenetic studies in the POHaD context. Most references cited by authors. In lines 126-145, examples refer to developmental programming effects in general, only.

9. In line 146, please define which RNAs? MicroRNAs?

10. Line 160, PDoH should be POHaD

11. Line 184. Please define what the "debate is about".

12. Lines 217-228, please introduve references in which the text is based on.

13. Line 258. Please correct PODaH
